# Fire Resistance of Fire-Protected Reinforced Concrete Beams Strengthened with Externally Bonded Reinforcement Carbon Fibre-Reinforced Polymers at the Full Utilisation Degree

**DOI:** 10.3390/ma16155234

**Published:** 2023-07-25

**Authors:** Piotr Turkowski

**Affiliations:** Instytut Techniki Budowlanej, Filtrowa 1, 00-611 Warszawa, Poland; p.turkowski@itb.pl; Tel.: +48-22-5664-169

**Keywords:** protection of concrete, heat transfer, structural design, carbon fibre-reinforced polymer, concrete beams, fire resistance

## Abstract

The fire resistance of reinforced concrete (RC) beams strengthened with externally bonded reinforcement carbon fibre-reinforced polymers (EBR CFRP) depends on the load level and degree of strengthening. Design guides for this type of structure recommend strengthening limits to prevent collapse of the structure due to vandalism, damage or fire. In practice, however, it is not uncommon to exceed the proposed limits. This necessitates protection of the CFRP adhesive against high temperatures, namely glass transition temperature levels. This paper presents the results of 10 full-scale experimental fire resistance tests of reinforced concrete beams with EBR CFRP strengthening, with and without fire protection and under various load levels including full utilisation. It was demonstrated that in cases where CFRP strengthening was not needed to carry load in an accidental fire scenario, test specimens with and without it achieved the same fire resistance. It was also proven that in cases where CFRP failure influenced the failure of the entire beam, fire protection was crucial for maintaining the beam’s load-bearing capacity and its thickness had to be substantial. A fire resistance rating of over 4 h was achieved, nevertheless.

## 1. Introduction

Commonly used design processes for reinforced concrete structural elements for accidental scenarios, such as fire, assume some level of load reduction (about 30% to 60%) [1]. This means that in a fire situation, FRP strengthening may no longer be required to carry loads if the strengthening level does not exceed that reduction. Neither the European fib Bulletin No. 14 of 2001 [2], the fib Bulletin No. 90 of 2019 [3], nor the U.S. ACI 440.2R-17 guide of 2017 [4] encourage designs where load-bearing capacity in the event of FRP strengthening failure due to mechanical damage, vandalism or fire is maintained. In practice, the fire resistance of elements designed in such a manner is related to the loss of strength of the reinforcing steel and concrete, and not the polymer composite. In such cases, fire protection is either not needed at all or is selected with the same criteria as for non-strengthened reinforced concrete elements.

In 2015, Firmo, Correia and Bisby [5] reviewed state-of-the-art FRP-strengthening methods of reinforced concrete structures, presenting, among other information, a 25-year history of fire resistance tests in the field [6,7,8,9,10,11,12]. The authors of the tests did not always report complete information about the construction of the test elements and the test conditions, but one feature is noted that unites all the tests carried out to date, namely that the fire resistance period is always greater than the time at which the FRP reinforcement failed. This means that the authors loaded the test elements below their load capacity prior to strengthening, which is consistent with the aforementioned design recommendations (see Table 1).

Similarly, in the case of concrete columns strengthened with FRP investigated experimentally [13,14,15], a satisfactory level of fire resistance was achieved (over 4 h) when fire insulation was used. Nevertheless, the temperature of glass transition was achieved at the surface of concrete to FRP. Such fire resistance did not result from the effectiveness of the FRP system, but from the presence of thermal insulation which protected the steel and concrete from excessive heating.

During that period, several enquiries were received at the Building Research Institute (ITB) concerning structures in which FRP strengthening was applied disregarding the limits proposed by [2,3,4]. One such building was a Warsaw office centre erected in 2011. As a result of execution errors, i.e., improper service interruption of concreting, one floor and its beams separated from the rest of the structure. This is not an uncommon error [16]. Technical examination of those elements showed the ability of the structure to carry dead weight, but the entire load that was imposed had to be carried by external bonded reinforcement. The required fire resistance class for these floors was still kept at REI 60, which meant that the EBR CFRP reinforcements had to be fireproofed in such a manner that they maintained their full strength during fire.

In order to address these problems, a series of tests were carried out at the Building Research Institute (ITB) between 2012 and 2016, involving regular and EBR CFRP-strengthened concrete beams, loaded both below and above their nominal load capacity before strengthening, with and without fire protection. This full range of large-scale tests allowed us to confirm an earlier conjecture about possible design cases and fill the research gap described by Harries [17] and Firmo [5]. For the first time, tests with a 100% degree of utilisation were performed on RC beams strengthened with EBR CFRP.

## 2. Materials and Methods

### 2.1. Test Specimens

Each beam was of the same size: 4700 mm length, 150 mm width and 450 mm height. Four 12 mm diameter bars (one bar in each corner) made of RB500W steel were used as longitudinal steel reinforcement in the beams. The lateral reinforcement consisted of 8 mm diameter steel stirrups, spaced at 20 cm, made of RB500W steel (Figure 1).

The concrete mixture was made with siliceous aggregate and the concrete class was C 16/20 (as per European standard [18]) for beams B-1 to B-6, and C 25/30 for beams B-7 to B-10. Beams B-1 to B-6 were conditioned for 120 days in ambient air temperature ranging from 10 °C to 30 °C and relative air humidity from 20% to 80% with 5% moisture content at the day of the test, while beams B-7 to B-10 were conditioned for 200 days in the same conditions, with 4% moisture content at the day of the test.

The upper surface of each beam was covered with 120 mm high and 600 mm wide autoclaved aerated concrete (AAC) blocks, in order to achieve three-sided heating.

CFRP strengthening was made with 4000 mm long, 50 mm wide and 1.2 mm thick strips of ultimate tensile strength of 3100 MPa and tensile elastic modulus of 170 GPa, externally bonded on the bottom beam surface, along the beams’ longitudinal axis, before load application, with a structural two component adhesive of glass transition temperature (*T*_g_) equal to 62 °C.

Two board fire protection systems were used:gypsum boards (GB) type GM-F, GM-H1, density of 850 kg/m^3^, thermal conductivity at 20 °C of 0.020 W/mK, single layer thickness of 25 mm;calcium silicate boards (CSB), density of 480 kg/m^3^, thermal conductivity at 20 °C of 0.009 W/mK, single layer thickness of 50 mm.

The fire protection scheme for the gypsum boards is given in Figure 2. In the case of the calcium silicate boards, either one or three layers of thermal insulation were used, of the same thickness at the bottom and on the sides of the beams.

The thermal properties of both materials, also used in the numerical analysis, are given in Figure 3 and Figure 4.

Prior to the fire tests, four mechanical tests were performed in order to obtain the value of the real load-bearing capacity of the beams before and after strengthening with CFRP strips. The values were also validated against calculations in accordance with fib Bulletins 14 and 90 [2,3] and software delivered by the FRP’s manufacturer. All values are given in Table 2.

The loading was applied with two hydraulic jacks, spaced at 130 ± 10 cm, to achieve uniform bending momentum over at least 25% of the beam’s support length.

An overview of the test specimen selection is given in Table 3 and Figure 5. Each beam was tested separately on the same test stand, between January and October 2015. The following groups of beams were distinguished:B-1 and B-10—beams with CFRP strengthening, without fire protection, loaded below their bending capacity before strengthening;B-2, B-3 and B-9—beams with and without CFRP strengthening, with fire protection, loaded below their bending capacity before strengthening;B-4 to B-8—beams with CFRP strengthening, with or without fire protection, loaded above their bending capacity before strengthening.

### 2.2. Test Method

Tests were conducted at the ITB Fire Testing Laboratory in Pionki. The K1 chamber of the CHIMERA furnace measured 4.30 m × 3.30 m × 3.70 m (*L* × *W* × *H*). It was made of a steel structure insulated with ceramic wool of low thermal conductivity and high surface emissivity and was equipped with ten gas burners located on shorter walls. Its maximum power was 6.2 MW.

The heating process was conducted according to the standard temperature/time curve as defined in EN 1363-1 [19], meeting the requirements concerning the parameters and tolerances of testing equipment, heating conditions, ambient temperature, furnace pressure and load, as defined therein. The use of furnaces for fire safety research along with considerations related to energy balance and heat transfer were given in [20].

Three main parameters were monitored during all tests:Fire protection failure time (*t*_FP_), defined as the time when the temperature rise on the bottom surface of the concrete beam exceeds 50 K/min. Such rate of temperature rise can be observed only after fire protection detachment or significant deterioration.EBR CFRP failure time (*t*_FRP_), defined as the time of noticeable detachment of CFRP strips or sudden, short-term increase in the rate of deflection, being the result of strain redistribution after CFRP failure.Beam failure time (*t*_R_), defined as the time when the rate of deflection of 5 mm/min was exceeded, calculated in accordance with EN 1363-1 [19].

The temperature of steel reinforcement and CFRP strengthening was also monitored with K-type thermocouples installed in the midspan of the beam and in additional bases under loading points and close to beam ends (see Figure 6).

## 3. Results

The recorded values of average temperature of the contact surface between the adhesive and the concrete (average value from thermocouples 3, 4 and 5 in bases A, B and C—see Figure 6) are given in Figure 7, the rate of deflection and deflection in Figure 8 and steel reinforcement temperature (medium value from thermocouples 1 and 2) in Figure 9. Values did not rise uniformly in all bases with significant differences between them, but over time they levelled out.

A summary of the results is given in Table 4.

This was dictated by the following reasons:leaking joints between fire boards, allowing hot gases from the furnace chamber to penetrate the air gap surrounding the CFRP strip; the effect was amplified by conditions in the furnace, i.e., imposed overpressure of 20 Pa at the level of the underside of the beam;the placement of thermocouples in the vicinity of the load application points, i.e., the points with the highest possible deflection and in the vicinity of vertical cracks;other minor effects such as: outflow of water from the concrete, geometric imperfections of the fire protection boards and uneven adhesive thicknesses.

where:

*t*_FRP_—time of CFRP strengthening system failure

*t*_FP_—time of fire protection system failure

*t*(d*D*/d*t*_lim_)—time to reach limiting rate of deflection

*t*_R_—fire resistance time (beam failure)

*T*_rf_(*t*_FRP_)—steel reinforcement temperature at the time of CFRP system failure

*T*_rf_(*t*_R_)—steel reinforcement temperature at the time of beam failure

*D*(*t*_FRP_)—deflection at the moment of CFRP system failure

*D*(*t*_FP_)—deflection at the moment of fire protection system failure

## 4. Discussion

The results from the tests on beams B-1 and B-10, both without fire protection, CFRP-strengthened and loaded below their bending capacity before strengthening, proved that a CFRP strip fully exposed to fire, fails almost instantly and detaches within a period of about 2 min from the start of heating. It was also demonstrated that when CFRP strengthening fails, the beam immediately shows a short-term increase in the rate of deflection but maintains fire resistance within set limits at the level expected for a regular reinforced concrete beam.

The investigations carried out on the next beam group B-2, B-3 and B-9 also confirmed the abovementioned conclusions. Particularly meaningful was the comparison of the time of fire protection failure, fire resistance time (time to reach the limiting rate of deflection) and steel reinforcement temperature in beams B-2 and B-3, which differed only in the presence of CFRP strengthening. In both cases, almost identical values were obtained, which explicitly demonstrates the lack of influence of the composite on the fire resistance of the element when the load does not exceed its load capacity before strengthening.

It was also noted that the failure of the fire-protected CFRP strips sheathed with fireproofing occurred when the glass transition temperature of the adhesive was exceeded, and the deflection of the beam was approximately 1/300 of the span, c. 10 times less than the limiting deflection value.

The tests described so far are consistent with those carried out by [6,7,8,9,10,11,12] (i.e., tests that followed the recommendations of fib and ACI [2,3,4], but which do not exhaust the applications of EBR FRP reinforcement that occur in construction practice).

The next group of beams, B-5 and B-6, fire-protected with gypsum boards, and B-7 and B-8, fire-protected with calcium silicate boards, as well as one beam without protection (B-4), constituted an entirely new research area. The loading of the beams in the tests utilised the full design bending capacity under normal conditions of these beams after strengthening. The author is not aware of any tests carried out in such an arrangement.

In order to confirm the actual role of the CFRP reinforcement experimentally, beam B-4 was tested without any protection. After just 1.07 min, the average temperature of the adhesive exceeded its glass transition temperature, reaching 70.4 °C, which led to rapid detachment of the composite and loss of the fire resistance of the beam. The other beams of this group behaved similarly, i.e., the loss of fire resistance occurred at the same time as the CFRP strip detachment, which in turn occurred when the glass transition temperature of the adhesive was exceeded (see Figure 10).

It was also found that where the heating time was longer (the longest was 289.50 min), the average temperature of the CFRP adhesive recorded at the time of composite failure was accordingly lower. This, however, was always above the manufacturer’s declared value of glass transition temperature. The increase in time taken for the CFRP to fail corresponded with a greater deflection of the beam when it occurred, with the highest recorded value being 12.5 mm (beam B-6).

The steel reinforcement shielded by the additional concrete cover, in comparison to the CFRP reinforcement, heated very little; thus, the temperature growth within that range did not significantly alter the mechanical properties of the steel.

A comparison of the temperature values calculated in the numerical analyses of beam BN-1 with the values recorded in tests on beams B-1, B-4 and B-10 indicates the correctness of the numerical model. The slight differences in the reinforcement temperature diagrams are most likely due to the actual deeper location of the measurement points in the beams. It should be borne in mind that, despite one’s best efforts, the thermocouples may shift during installation and vibration of the concrete. Conversely, higher temperature values than in numerical calculations for directly heated surfaces, in this case the underside surface of the concrete beam, are typical of fire resistance tests and measurements made with thermocouples on concrete surfaces. The measuring junction of the thermocouple is actually slightly farther away from the heated concrete surface and measures the temperature of the gases and the radiation of the furnace baffles, rather than the temperature of the concrete [21].

The results for the BN-2 and BN-3 models, simulating heat flow for beams with GB insulation, are in good agreement with experimental data, especially in terms of temperature values for steel reinforcement in the B-2 and B-3 beams, and the temperature of CFRP in beam B-6. The calculated temperature values are always conservative in relation to the values recorded in the fire resistance tests during the period in which the fire protection retained its adhesion to the beam.

Quite the opposite can be noted for the BN-4 and BN-5 models, where calculated steel reinforcement values were overestimated and concrete surface temperature values were underestimated. The numerically determined time of CFRP system failure was 47% higher than that derived from the fire resistance test for 150 mm insulation thickness, and 22% lower for an insulation thickness of 50 mm.

## 5. Conclusions

The experimental investigations and numerical analyses presented in this study enabled the formulation of the following conclusions:It is possible to fireproof reinforced concrete beams with EBR-CFRP, loaded above their load-bearing capacity before strengthening, and achieve more than 4 h of fire resistance. This has not yet been demonstrated in actual full-scale tests.Regardless of heating time (in the range from 1 to 300 min), failure of the EBR CFRP strengthening system was caused by achieving and exceeding the glass transition temperature of the adhesive, resulting in complete and rapid detachment of the CFRP. This resulted in the immediate achievement of fire resistance criteria (rate of deflection and, in many cases, collapse of the beam), but only for beams loaded over their load-bearing capacity prior to strengthening. In other cases, only a temporary growth of deflections was noted at that moment, with no influence on the further behaviour of the element.Until strengthening system failure, the beams demonstrated a small growth in deflections (about 5 mm, which is about 2% of the limiting value for the analysed experimental structure [19]). Also, higher values of deflections were registered for longer heating periods.

It is also worth mentioning, that the registered adhesive temperature at the moment of CFRP failure for the beam heated the longest (289 min) was lower than for the beams with shorter heating periods. This could indicate that prolonged exposure may result in lower glass transition temperature, but this claim requires further research.

The practical significance of the research presented here is to prove the feasibility of expanding the application of EBR CFRP strengthening on RC elements whose load capacity in fire conditions is fully dependent on them. Vandalism or mechanical damage make the ACI and fib recommendations still valid, but a fire scenario may be managed with proper fire protection.

## Figures and Tables

**Figure 1 materials-16-05234-f001:**
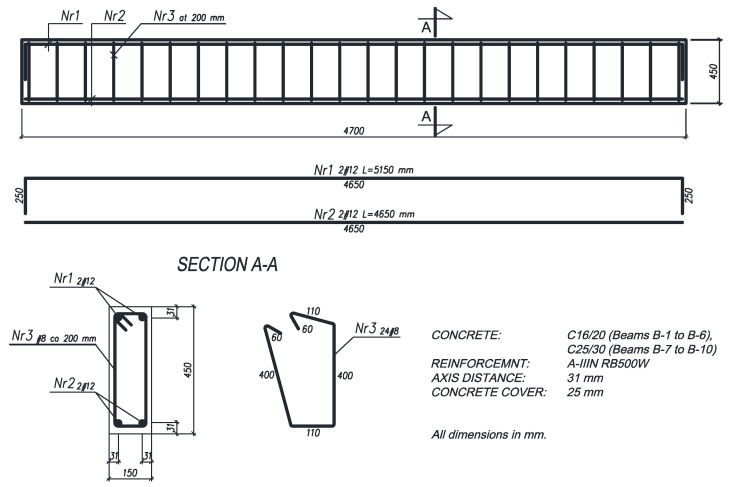
Reinforced concrete beam scheme.

**Figure 2 materials-16-05234-f002:**
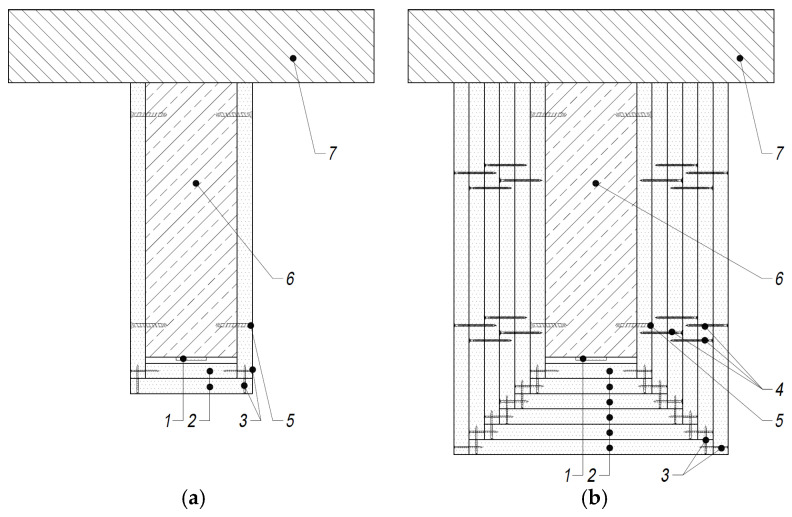
Gypsum board fire protection system: (**a**) thickness—25/50 mm; (**b**) thickness—150 mm. 1—CFRP strip 50 mm × 1.2 mm; 2—gypsum board 25 mm; 3—steel screw 3.5 mm × 50 mm at 150 mm; 4—steel screw 3.5 mm × 70 mm at 150 mm; 5—steel screw for concrete at 500 mm; 6—reinforced concrete beam 450 mm × 150 mm; 7—AAC topping 150 mm.

**Figure 3 materials-16-05234-f003:**
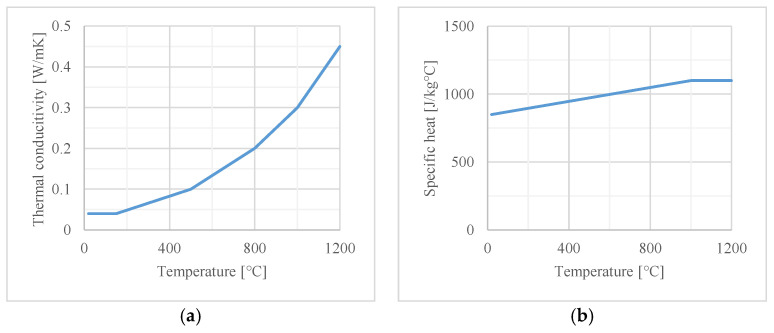
Thermal properties of calcium silicate boards with density of 480 kg/m^3^: (**a**) thermal conductivity; (**b**) specific heat.

**Figure 4 materials-16-05234-f004:**
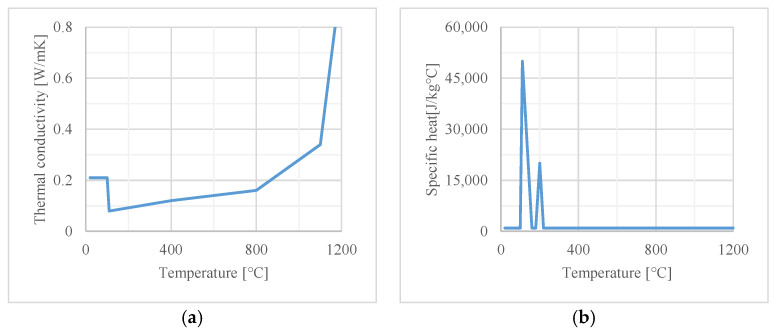
Thermal properties of gypsum boards with density of 850 kg/m^3^: (**a**) thermal conductivity; (**b**) specific heat.

**Figure 5 materials-16-05234-f005:**
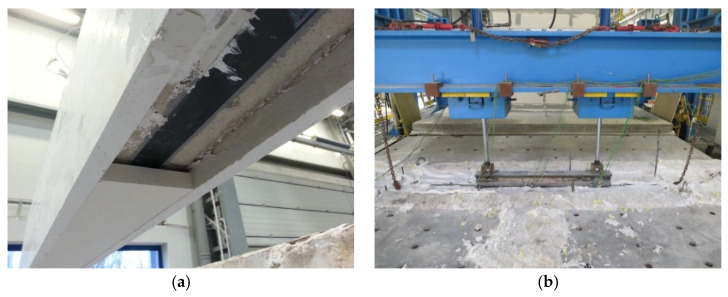
Test specimen prior to the test: (**a**) assembly; (**b**) loading system; (**c**) B-2; (**d**) B-1.

**Figure 6 materials-16-05234-f006:**
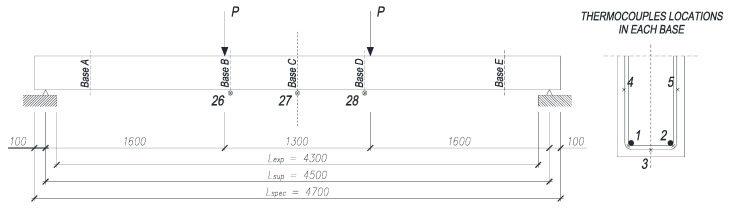
Thermocouple location in test specimen and loading points. 1 to 5, 26, 27, 28—thermocouple number; P—force applied.

**Figure 7 materials-16-05234-f007:**
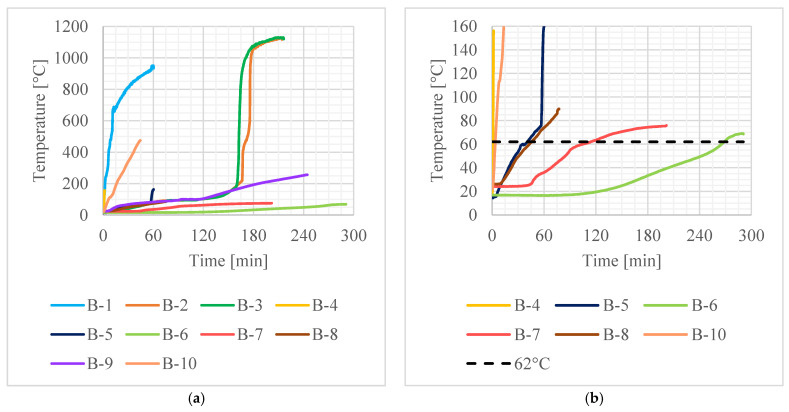
Average temperature of the CFRP strip during tests: (**a**) all data; (**b**) detailed view.

**Figure 8 materials-16-05234-f008:**
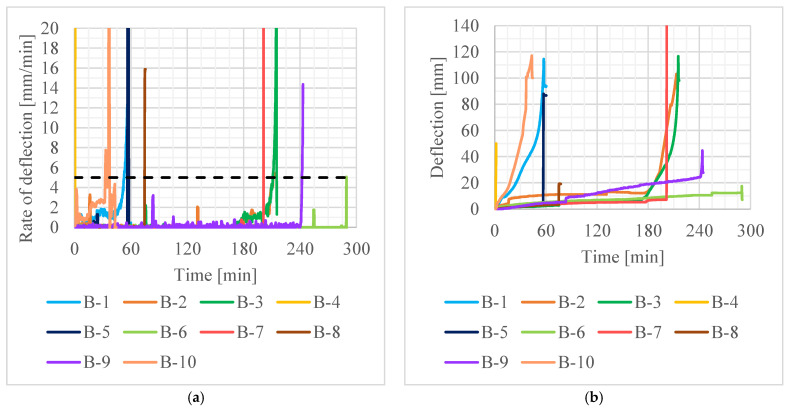
Deformation of beams in tests: (**a**) rate of deflection; (**b**) deflection.

**Figure 9 materials-16-05234-f009:**
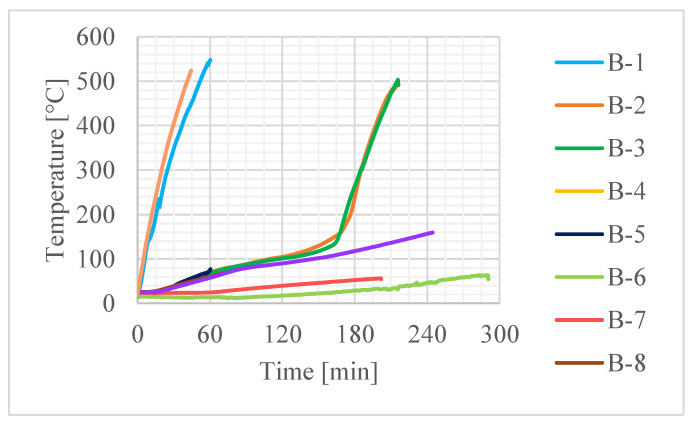
Steel reinforcement temperature of beams in tests.

**Figure 10 materials-16-05234-f010:**
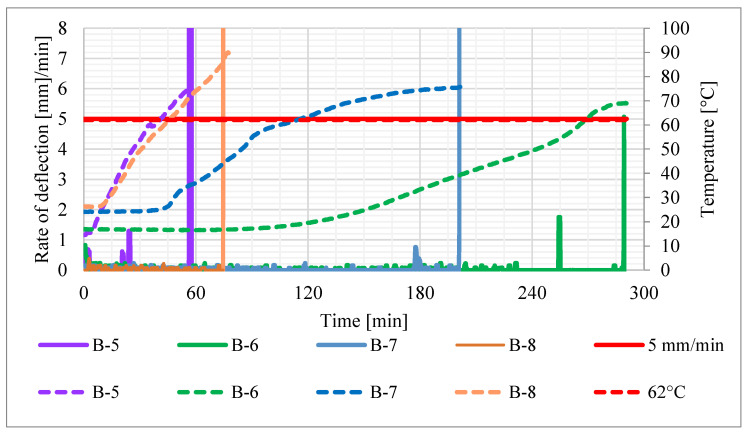
Rate of deflection (continuous line) vs. CFRP temperature (dashed line).

**Table 1 materials-16-05234-t001:** Summary of past fire resistance tests of RC beams strengthened with EBR CFRP.

Authors	Strength Increase [%]	Applied Load/Ultimate Capacity [%]	CFRP Failure Time [min]	Fire Resistance [min]
Deuring [6]	n.a.	55	62–65	>120
Blontrock et al. [7]	55	40	7–39	>90
Williams et al. [8]	15	48	16–57	>240
Adelzadeh et al. [9]	n.a.	71	30	>240
Ahmed and Kodur [10]	50	50	20–25	>180
Firmo et al. [11]	94	47	23–167	>210
Firmo and Correia [12]	74	36	2–70	>2–70

n.a.: Not available.

**Table 2 materials-16-05234-t002:** Load-bearing moment values calculated or experimentally investigated for beams.

Parameter	Designation	Value [kNm]
Design load-bearing capacity in normal conditions, before strengthening	*M* _Rd,RC_	36.8
Design load-bearing capacity in fire conditions *t* = 0, before strengthening	*M* _Rd,0,fi,RC_	43.8
Load-bearing capacity measured in strength test, before strengthening	*M* _Rd,RC_	55.0
Design load-bearing capacity in normal conditions, after strengthening	*M* _Rd,CFRP_	71.1
Load-bearing capacity measured in strength test, after strengthening	*M* _R,CFRP_	81.0
Design load-bearing capacity in normal conditions, before strengthening	*M* _Rd,RC_	36.8
Design load-bearing capacity in fire conditions *t* = 0, before strengthening	*M* _Rd,0,fi,RC_	43.8

**Table 3 materials-16-05234-t003:** Beams tested in the experimental part of this study.

Beam No.	Strengthening	Fire Protection	Bending Moment in Test [kNm]	Applied Load/Ultimate Capacity [%]
B-1	CFRP 50 × 1.2 mm	none	44	62
B-2	none	GB, 25 mm	44	100
B-3	CFRP 50 × 1.2 mm	GB, 25 mm	44	62
B-4	CFRP 50 × 1.2 mm	none	71	100
B-5	CFRP 50 × 1.2 mm	GB, 25 mm	71	100
B-6	CFRP 50 × 1.2 mm	GB, 150 mm	71	100
B-7	CFRP 50 × 1.2 mm	CSB, 150 mm	71	100
B-8	CFRP 50 × 1.2 mm	CSB, 50 mm	71	100
B-9	CFRP 50 × 1.2 mm	CSB, 50 mm	56	79
B-10	CFRP 50 × 1.2 mm	none	46	65

**Table 4 materials-16-05234-t004:** Results from fire resistance tests of the beams.

Beam No.	*t*_FRP_ [min]	*t*_FP_ [min]	*t*(d*D*/d*t*_lim_) = *t*_R_ [min]	*T*_rf_(*t*_R_) [°C]	*T*_FRP_(*t*_FRP_) [°C]	*T*_FRP_(*t*_R_) [°C]	*D*(*t*_FRP_) [mm]	*D*(*t*_FP_) [mm]
B-1	2.25	n.a.	52.75	506.0	196.5	911.1	0.2	n.a.
B-2	n.a.	168.00	211.50	477.9	n.a.	n.a.	n.a.	12.2
B-3	75.50	160.00	211.00	478.4	88.9	1127.9	2.8	6.1
B-4	1.07	n.a.	1.07	16.9	70.4	70.4	0.3	n.a.
B-5	56.50	56.50 ^1^	56.50	68.4	75.4	75.4	3.4	3.4 ^2^
B-6	289.50	289.50 ^1^	289.50	62.5	68.9	68.9	12.5	12.5 ^2^
B-7	201.75	201.75 ^1^	201.75	56.2	75.2	75.2	7.1	7.1 ^2^
B-8	75.00	75.00 ^1^	75.00	70.1	85.8	85.8	3.4	3.4 ^2^
B-9	83.50	242.0 ^1^	242.0	157.9	90.9	254.4	5.3	26.8 ^2^
B-10	2.25	n.a.	32.75	428.5	40.3	363.6	1.3	n.a.

^1^ value not reached—time to reach load-bearing capacity is given. ^2^ value not reached—deflection at the moment of reaching the load-bearing capacity is given. n.a.: Not applicable.

## Data Availability

Not applicable.

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
