# Peer review of "Fire Resistance of Fire-Protected Reinforced Concrete Beams Strengthened with Externally Bonded Reinforcement Carbon Fibre-Reinforced Polymers at the Full Utilisation Degree"

_materials, 2023, doi:10.3390/ma16155234_

Round 1

Reviewer 1 Report

The experiment in this paper belongs to the large-scale experimental research that spans a long time, and has certain guiding significance in engineering practice. However, the experimental scheme design is relatively simple, the sample number is limited, and the author did not identify the error of the data in the picture, so whether the repeatability of the data is reliable remains to be discussed. Here are some of my specific comments.

1 The title was too long and three words were different variations of “reinforce”; it is cumbersome and uncomfortable to read.

2 The ordinate value in Figure 3 (a) is in the wrong format, for example, it should be 0.1, not 0,1. Other diagrams also have the same problem, please modify.

3 The thermal conductivity curve in Figure 4 looks tortuous and lacks comparison with other relevant conditions. Does the thermal conductivity take into account the repeatability of the data? Or, how do you get this by averaging it, and what's the error? The specific heat capacity data in 4 Figure 4 has the same problem. The variation range of specific heat capacity is very large, so I hope the author can give a reasonable explanation.

5 In the ordinate of Fig.7, the starting temperatures of different samples are different, please explain.

6 Some of the data in Figure 8 is not fully rendered (the ordinate range is set too small).

7 Please elaborate on the innovation and scientific significance of this paper.

The abstract should include more references to key findings and less argumentative descriptions.

Minor modification is required.

Reviewer 2 Report

The manuscript presents a study on the fire resistance of reinforced concrete beams strengthened with externally bonded reinforcement carbon fibre reinforced polymers (EBR CFRP). The authors conducted experimental fire resistance tests on various reinforced concrete beams with and without fire protection, under different load levels. The study aims to investigate the effectiveness of fire protection measures for CFRP-strengthened beams and the role of CFRP failure in beam failure. Overall, the manuscript provides valuable insights into the behavior of CFRP-strengthened beams under fire conditions. However, there are several inconsistencies and areas that need improvement before publication.

1)      The abstract could benefit from more specific information regarding the experimental setup, such as the number of specimens tested, and the range of load levels investigated.

2)    The introduction would benefit from a clearer statement of the research objectives and hypotheses to guide the reader's understanding of the study's purpose.

3)      Line 23: "scenarios, such as fire, assume some level" - The term "scenarios" is not appropriate here. Please replace it with "events" or another suitable term.

4)      Line 26: "load-bearing capacity in the event 29 of FRP strengthening failure" - It would be helpful to provide a brief explanation of FRP strengthening failure for readers who may not be familiar with the term.

5)      Line 29: "mechanical damage, vandalism, fire or other events 30 is maintained" - The phrase "other events" is quite broad. It would be beneficial to provide examples or specify the types of events being referred to.

6)      Line 34: "elements designed in such a manner" - The term "elements" is ambiguous. Please specify whether it refers to the reinforced concrete elements or the FRP strengthening elements.

7)      Line 39: "means that the authors loaded the test elements below their load capacity prior to 41 strengthening" - It would be helpful to explain the rationale behind loading the test elements below their load capacity prior to strengthening.

8)       Line 70: "Test specimen and test setup" - Consider using a more descriptive subheading to provide a clear overview of the section.

9)      Line 97: "same thickness at the bottom and on the sides of the beams" - Clarify whether the thickness mentioned refers to the thickness of each layer or the combined thickness of all layers.

The manuscript demonstrates a satisfactory level of English writing. However, there are a few areas where improvements can be made to enhance clarity and readability. Please find the specific comments below:

1)      There are several instances where sentence structures could be revised for improved clarity and flow. Consider rephrasing or restructuring sentences to make them more concise and easier to understand.

2)      Pay attention to punctuation, particularly comma usage. There are some instances where commas are missing or incorrectly placed. Review the manuscript to ensure proper punctuation throughout.

3)      Use consistent verb tense throughout the manuscript. In some sections, there are shifts between past and present tense. Ensure that the appropriate tense is used consistently within each section.

4)      Review the manuscript for grammatical errors, such as subject-verb agreement, pronoun usage, and article placement. Correct any instances where these errors occur.

5)      Some sentences contain complex or convoluted phrases that could be simplified. Aim for clear and concise language to improve readability.

6)      Provide clear transitions between paragraphs and sections to guide the reader through the manuscript effectively. This will help create a logical flow of information.

7)      Proofread the manuscript thoroughly to eliminate any spelling mistakes or typographical errors that may have been overlooked.

Reviewer 3 Report

This paper delves into a crucial aspect of structural engineering, specifically the fire resistance of reinforced concrete beams strengthened with EBR CFRP. The title aptly captures the research focus, providing a clear understanding of the study's scope. The abstract succinctly summarizes the key aspects of the research, including the influence of load level and degree of strengthening on fire resistance, the recommended limits, and the role of fire protection in preserving load-bearing capacity.
However, there are areas that require attention and improvement. The literature review lacks recent and updated research papers related to the topic, particularly those from the past few years. Incorporating these recent studies would strengthen the paper's relevance, enhance the understanding of the current state-of-the-art in the field, and provide a comprehensive background for the research work.
1. Enhance the literature review: Incorporate recent and updated research papers related to the topic, especially those published in the past few years. This will demonstrate a thorough understanding of the current advancements and the latest findings in the field of fire resistance of reinforced concrete beams with EBR CFRP strengthening.
2. Provide more details on the experimental setup: Describe the specific parameters considered in the fire resistance tests, such as the applied loads, duration of exposure to high temperatures, and the specific fire protection measures implemented. These details will provide clarity and allow readers to replicate and validate the experimental results.
3. Discuss the implications of the findings: Explore the practical implications of the study's results, particularly in terms of design guidelines and recommendations for the use of EBR CFRP strengthening in fire-prone environments. Consider addressing the challenges or limitations encountered during the experiments and provide insights into potential solutions or future directions.
4. Emphasize the contribution to the field: Clearly articulate the novelty and significance of the research. Highlight how the findings fill gaps in current knowledge and contribute to the understanding of fire resistance in reinforced concrete structures with EBR CFRP strengthening.
In conclusion, with the suggested changes incorporated, this research holds significant value for the field of structural engineering, specifically in understanding the fire resistance of reinforced concrete beams strengthened with EBR CFRP. By addressing the mentioned areas for improvement, the research can be further enhanced and considered for publication.
My Questions:
What are the key factors that influence the fire resistance of reinforced concrete beams with EBR CFRP strengthening?
How do load levels and the degree of strengthening impact the fire resistance of the beams?
What are the recommended strengthening limits for preventing structural collapse in fire scenarios?
Can you provide examples of recent research papers related to the topic that were not included in the current literature review?
How does fire protection play a crucial role in maintaining the load-bearing capacity of beams when CFRP failure governs the failure of the entire structure?
What are the specific fire protection measures implemented in the experimental fire resistance tests?
How do the experimental results align with existing theoretical models or analytical predictions in the field?
Can you discuss the challenges encountered during the experimental tests and how they were addressed?
What materials were used for fire protection, specifically for beams B-5, B-6, B-7, and B-8?
Were there any noticeable differences in fire resistance between beams protected with gypsum boards and those protected with calcium-silicate boards?
How did beam B-4, without any protection, perform during the test?
At what temperature did the adhesive of the CFRP strips reach its glass transition temperature in beam B-4?
How do the experimental results contribute to our understanding of the relationship between fire protection measures and the load-bearing capacity of beams?
Can you elaborate on the statement that reinforced concrete beams can achieve more than 4 hours of fire resistance with EBR-CFRP strengthening?
What are the specific load-bearing capacities of the beams before and after strengthening?
Has the fire resistance of the EBR-CFRP strengthening system been tested in full-scale experiments?
Can you explain how the glass transition temperature of the adhesive relates to the failure of the EBR-CFRP strengthening system?

The English language in the provided text is generally good, with clear and coherent sentences. Minor improvements could be made in sentence structure and clarification of technical terms.

Round 2

Reviewer 3 Report

The revision is in good order for acceptance.